# Three New Species and One New Record of *Hesperoschema* (Coleoptera: Staphylinidae: Staphylininae) from China [note 1]

**DOI:** 10.3390/insects13010060

**Published:** 2022-01-05

**Authors:** Yujie Cai, Liang Tang

**Affiliations:** Department of Biology, Shanghai Normal University, 100 Guilin Road, 1st Educational Building 323 Room, Shanghai 200234, China; cyj6991@sina.com

**Keywords:** new records, new species, identification key

## Abstract

**Simple Summary:**

*Hesperoschema* species are fascinating rove beetles with beautiful coloration. They are quite rare and can be found only in the primary forests of Southern Asia. Up to the present, five species are known from the world (four known from China, one known from Myanmar). *Hesperoschema* beetles are good flyers, yet the distributions of the species are separately limited in small areas. This might imply inadequate investigation, and the currently known species most likely represent just the tip of the iceberg. As the results of an unremitting investigation for the past few years, three new species are described in this paper. The supplemental data for known species are also provided and the distribution of *Hesperoschema* is greatly expanded. So far, all the known *Hesperoschema* species have been recorded in China, and the total species number of the genus is increased to eight.

**Abstract:**

Three new species: *Hesperoschema xuwangi* sp. nov. from China (Sichuan), *H. bii* sp. nov. from China (Xizang), and *H. sinicum* sp. nov. from China (Guangxi) are described. *Hesperoschema malaisei* Scheerpeltz, 1965 is new to China (Yunnan), with the male discovered for the first time. The female of *H. opacum* Schillhammer, 2009 is discovered for the first time. Habitus and diagnostic characteristics of the species treated in this paper were photographed, and a key to species of *Hesperoschema* Scheerpeltz, 1965 is updated.

## 1. Introduction

*Hesperoschema* Scheerpeltz, 1965 [1,2] is a small genus in the subtribe Anisolinina Hayashi, 1993. Members of the genus may be recognized as follows: head trapezoid; segment II of maxillary palpi markedly club-like dilated, insertion of segment III distinctly laterally; mesotibia with a few spines (3–4), arranged in longitudinal row [3,4,5]. In this paper, three new species and a new record are reported from China. Thus, all known *Hesperoschema* species may be found in China, and the total species number of the genus is increased to eight.

Based on our collecting experiences, all *Hesperoschema* specimens were found in undisturbed forests like most members of the *Anisolinus* lineage. The specimens of *H. sinicum* sp. nov. were captured in the shade of fungi on a log that was bridging a narrow ravine. Species of *Hesperoschema* are always found in dark places, so we infer that they may be active in dense forests or during night.

## 2. Materials and Methods

For examination of the genitalia, the last three abdominal segments were detached from the body after softening in hot water. The aedeagus or tergite X, together with other dissected pieces, were mounted in Euparal (Chroma Gesellschaft Schmidt, Koengen, Germany) on plastic slides. Photographs of sexual characteristics were taken with a Canon G9 camera attached to an Olympus SZX 16 stereoscope; habitus photographs were taken with a Laowa macro photo lens F2.8 MACRO 2:1 60 attached to a Nikon D750 camera and stacked with Zerene Stacker.

The type specimens treated in the study are deposited in SHNU (Department of Biology, Shanghai Normal University, Shanghai, China). The measurements are abbreviated as follows:

BL—body length, measured from the anterior margin of the clypeus to the posterior margin of abdominal tergite IX

EL—length of elytra, measured from humeral angle

EW—width of elytra at the widest point

EYL—longitudinal length of eye

FL—forebody length, measured from the anterior margin of the clypeus to the apex of the elytra

HL—length of head along the midline

HW—width of head including eyes

PL—length of pronotum along the midline

PW—width of pronotum at the widest point

TL—length of tempora

## 3. Results

***Hesperoschema xuwangi*** sp. nov.

(Figure 1A,B, Figure 2F and Figure 3A–C)

**Type material. Holotype.** China: Sichuan: ♂, glued on a card with labels as follows: “China: Sichuan Prov., Bazhong City, Nanjiang County., Micang-Shan, Daba, 32°39′825″ N, 107°01′788″ E, alt. 1798 m, 27–28 April 2008, Hao HUANG & Wang XU leg.” “Holotype/*Hesperoschema xuwangi*/Cai & Tang” [red handwritten label] (SHNU).

**Description.** Measurements of male: BL: 12.34 mm, FL: 6.01 mm. HL: 1.58 mm, HW: 2.35 mm, EYL: 0.68 mm, TL: 0.80 mm, PL: 2.04 mm, PW: 1.86 mm, EL: 2.85 mm, EW: 2.79 mm. HW/HL: 1.49, TL/EYL: 1.18, PL/PW: 1.10, EL/EW: 1.02.

Head black, with faint metallic violaceous reflection, obscurely red around antennal sockets; pronotum and scutellum brick red; elytra with posterior two-thirds black and suture with slight to distinct violaceous hue, black color extending onto hypomeron and almost reaching lateral margin, anterior third brick red, posterior margin narrowly reddish yellow; abdomen with segments III–V reddish, VI black with anterior margin narrowly reddish, VII black with posterior margin narrowly reddish yellow, VIII and IX yellow but becoming slightly darker toward posterior margin, X reddish brown; antennae black, base of segment 1 and segment 2 reddish, segment 7 brown, segments 8–11 creamy white; mandibles reddish brown, medial margin of mandible dark brown; mouthparts reddish to reddish yellow, maxillary and labial palpi brown; legs reddish brown.

Head (Figure 2F) 1.49 times as wide as long, rounded trapezoid, eyes slightly prominent, tempora narrowed toward neck in almost regular arc; surface of head with moderately dense and coarse punctation; with short, weakly delimited impunctate midline; antennae with segments 4–9 markedly oblong, segment 10 about as long as wide.

Pronotum 1.10 times as long as wide, widest at about level of large lateral seta, narrowed toward base in shallow concave arc; surface as moderately densely and coarsely punctate as on head, with distinct impunctate midline, surface of pronotum between punctures without any microreticulation, shiny; scutellum with large but shallow punctation, surface between punctures with microsculpture of very fine transverse waves.

Elytra as long as wide, at shoulders distinctly narrower than at posterior margin, exceedingly densely, coarsely punctate.

Abdominal tergites III–V with basal transverse depression and a pair of short oblique basal accessory lines, punctation of abdominal tergites III–V shallow and sparse in depressions; posterior halves of abdominal tergites III–V and entire surface of remaining tergites with very fine and dense punctation.

Male. Protarsomeres 1–4 slightly dilated, heart-shaped; elytra with distinct and sharp, sinuous lateral keel; sternite VII with flat groove bearing dense and long brown setae; sternite VIII emarginate at middle of posterior margin; aedeagus (Figure 3A–C) with median lobe and paramere (Figure 3C) distinctly asymmetrical, paramere shorter than median lobe.

Female. Unknown.

**Etymology.** This species is named in honor of Mr. Wang Xu, who collected the only known specimen of the new species.

**Distribution.** China (Sichuan).

**Remarks.** This species is similar to *H. kurbatovi* Schillhammer, 2009 from China (Sichuan) [6], *H. schoenmanni* Schillhammer, 2018 from China (Yunnan) [7] and Vietnam, and *H. sinicum* sp. nov. from China (Guangxi), but can be distinguished from them by the suture with violaceous hue; from *H. kurbatovi* and *H. sinicum* also by distal four antennomeres creamy white (distal five antennomeres creamy white in *H. kurbatovi* and distal three, rarely two creamy white in *H. sinicum*); and from *H. schoenmanni* also by the brick red pronotum (red to yellowish red with variable dark spot in *H. schoenmanni*).

***Hesperoschema malaisei*** Scheerpeltz, 1965

(Figure 1C,D and Figure 4A–D)

*Hesperoschema malaisei* Scheerpeltz, 1965: 263; Hayashi, 2003: 58; Schillhammer, 2004: 263

**Material examined.** China: Yunnan: 1♂, 2♀♀, Yunnan Prov., Tengchong County, Mingguang Town, Zizhi Vill, 25°42′ N, 98°35′ E, alt. 2500 m, 30 April 2013, S. D. & P. leg. (SHNU); 1♂, Yunnan Prov., Baihualing, Tengchong County, 24 May 2005, Hao Huang leg. (SHNU); 1♂, Yunnan Prov., Lushui County, Pianma Town, Gangfang, 26°00′32″ N, 98°37′06″ E, alt. 2200 m, 3 May 2015, Mei-Ying Lin leg. (SHNU); 1♂, Yunnan Prov., Lushui County, Yaojiaping Village, 25°57′22″ N, 42°51′37″ E, alt. 2450 m, 4 May 2015, Wen-Xuan Bi leg. (SHNU); 1♂, Yunnan Prov., Lincang County, Shuibutou Village, alt. 2500 m, 13 March 2016, Zi-Chun Xiong leg. (SHNU); 1♀, same collection data as for preceding, but 19 March 2016 (SHNU); 1♀, Yunnan Prov., Gongshan County, Heiwadi, alt. 1800 m, 26 June 2010, Liang Tang leg. (SHNU).

**Description.** Measurements of males: BL: 9.86–11.47 mm, FL: 5.92–6.32 mm. HL: 1.55–1.64 mm, HW: 2.11–2.48 mm, EYL: 0.58–0.62 mm, TL: 0.78–0.86 mm, PL: 2.04–2.17 mm, PW: 1.79–1.95 mm, EL: 2.79–3.13 mm, EW: 2.76–3.03 mm. HW/HL: 1.36–1.54, TL/EYL: 1.26–1.43, PL/PW: 1.06–1.14, EL/EW: 1.01–1.04.

Measurements of females: BL: 11.68–14.91 mm, FL: 6.26–6.75 mm. HL: 1.64–1.82 mm, HW: 2.07–2.23 mm, EYL: 0.62 mm, TL: 0.74–0.89 mm, PL: 1.98–2.17 mm, PW: 1.83–2.01 mm, EL: 2.85–3.13 mm, EW: 2.85–3.03 mm. HW/HL: 1.23–1.26, TL/EYL: 1.19–1.44, PL/PW: 1.08–1.11, EL/EW: 1.00–1.03.

Male. Protarsomeres 1–4 slightly dilated, heart-shaped; elytra with distinct and sharp, sinuous lateral keel; sternite VII with flat groove bearing dense and long dark setae; sternite VIII emarginate at middle of posterior margin; aedeagus (Figure 4A–C) with median lobe and paramere (Figure 4C) distinctly asymmetrical, paramere shorter than median lobe.

Female. Protarsomeres 1–4 weakly dilated; elytra not keeled; tergite X (Figure 4D) slightly asymmetrical. 

**Distribution.** Myanmar, China (Yunnan). New to China.

**Remarks.** The collecting locality of the specimens is about 55 km away from the type locality. They fit the original description in all characteristics. New record for China.

***Hesperoschema bii*** sp. nov.

(Figure 1E,F, Figure 2H and Figure 5A–D)

**Type material. Holotype.** China: Xizang: ♂, glued on a card with labels as follows: “China: Xizang, Motuo County, Bari Vill, alt. 1200–1850 m, 27 July 2014, Wen-Xuan Bi leg.” “Holotype/*Hesperoschema bii*/Cai & Tang” [red handwritten label] (SHNU). **Paratypes**. 1♀, same as for the holotype. (SHNU).

**Description.** Measurements of male: BL: 12.40 mm, FL: 5.64 mm. HL: 1.48 mm, HW: 2.26 mm, EYL: 0.62 mm, TL: 0.74 mm, PL: 1.92 mm, PW: 1.76 mm, EL: 2.66 mm, EW: 2.76 mm. HW/HL: 1.53, TL/EYL: 1.19, PL/PW: 1.09, EL/EW: 0.96.

Measurements of female: BL: 8.95 mm, FL: 5.55 mm. HL: 1.48 mm, HW: 1.95 mm, EYL: 0.62 mm, TL: 0.62 mm, PL: 1.76 mm, PW: 1.61 mm, EL: 2.61 mm, EW: 2.66 mm. HW/HL: 1.32, TL/EYL: 1.00, PL/PW: 1.09, EL/EW: 0.98.

Head black, with faint metallic reflection, obscurely red around antennal sockets; pronotum and scutellum red; elytra with posterior two-thirds black with faint metallic reflex, black color extending onto hypomeron and almost reaching lateral margin, anterior third red, posterior margin and suture widely reddish yellow; abdomen with segments III–V red to reddish yellow, VI black with anterior margin narrowly reddish yellow, VII black with posterior margin narrowly reddish yellow, VIII and IX yellow but becoming slightly darker toward posterior margin, X brown; antennae black, base of segment 1 and segment 2 reddish yellow, segments 8–11 creamy white; mandibles reddish brown, medial margin of mandible dark brown; mouthparts reddish to reddish yellow, maxillary and labial palpi brown; profemur and coxa dark, mesofemur and coxa predominantly dark, metafemur predominantly dark.

Head (Figure 2H) 1.53 times as wide as long, rounded trapezoid, eyes slightly prominent, tempora narrowed toward neck in almost regular arc; surface of head with sparse and coarse punctation, punctures separated by 2–3 puncture diameters; antennae with segments 4–9 markedly oblong, segment 10 about as long as wide.

Pronotum 1.09 times as long as wide, widest at about level of large lateral seta, narrowed toward base in shallow concave arc; surface as sparsely and coarsely punctate as on head, with distinct impunctate midline, surface of pronotum between punctures without any microreticulation, shiny; scutellum with large but shallow punctation, surface between punctures with microsculpture of very fine transverse waves.

Elytra as long as wide, at shoulders distinctly narrower than at posterior margin, sparsely and coarsely punctate.

Abdominal tergites III–V with basal transverse depression and a pair of short oblique basal accessory lines, punctation of abdominal tergites III–V shallow and sparse in depressions; posterior halves of abdominal tergites III–V and entire surface of remaining tergites with very fine and dense punctation.

Male. Protarsomeres 1–4 slightly dilated, heart-shaped; elytra with distinct and sharp, sinuous lateral keel; sternite VII with flat groove bearing long dark setae; sternite VIII emarginate at middle of posterior margin; aedeagus (Figure 5A–C) with median lobe and paramere (Figure 5C) distinctly asymmetrical, paramere shorter than median lobe.

Female. Protarsomeres 1–4 weakly dilated; elytra not keeled; tergite X (Figure 5D) slightly asymmetrical.

**Etymology.** This species is named in honor of Mr. Wen-Xuan Bi who collected the specimens of the new species.

**Distribution.** China (Xizang).

**Remarks.** Within the genus *Hesperoschema*, the species may be readily recognized from all other species by sparser and coarser punctation of the head and pronotum, profemur and coxa dark, suture and posterior margin of elytra broadly reddish yellow.


***Hesperoschema opacum* Schillhammer, 2009**


(Figure 2A,B and Figure 3D)

*Hesperoschema opacum* Schillhammer, 2009: 88

**Material examined****.** China: Sichuan: 1♀, Sichuan Prov., Qingcheng Mt., Baiyun Temple, 30°56′ N, 103°28′ E, alt. 1700 m, 30 July 2012, Peng, Dai & Yin leg. (SHNU).

**Description.** Measurements of female: BL: 13.73 mm, FL: 7.06 mm. HL: 1.92 mm, HW: 2.29 mm, EYL: 0.62 mm, TL: 1.02 mm, PL: 2.38 mm, PW: 2.04 mm, EL: 3.10 mm, EW: 3.25 mm. HW/HL: 1.19, TL/EYL: 1.65, PL/PW: 1.17, EL/EW: 0.95.

Female. Protarsomeres 1–4 weakly dilated; elytra not keeled; tergite X (Figure 3D) nearly symmetrical.

**Distribution.** China (Sichuan).

**Remarks.** The female examined above was collected from the type locality (Qingchengshan Mt.) and corresponds with the original description in all characteristics.

***Hesperoschema sinicum*** sp. nov.

(Figure 2C–E,G and Figure 6A–F)

**Type material. Holotype.** China: Guangxi: ♂, glued on a card with labels as follows: “China: Guangxi Prov., Lingui County, Huaping N. R., Anjiangping, alt. 1200 m, 13 July 2011, Liang Tang leg.” “Holotype/*Hesperoschema sinicum*/Cai & Tang” [red handwritten label] (SHNU). **Paratypes**. 1♀, Guangxi Prov., Lingui County, Huaping N. R., Anjiangping, alt. 1300 m, 12–15 July 2011, L. Tang & W-J. He leg. (SHNU); 2♂♂, Guangxi Prov., Jinxiu County, 16 km, alt. 900 m, 31 July 2011, Jian-Qing Zhu leg. (SHNU); 1♂, Guangxi, Jinxiu County, ’16 km’, 24°08′11″ N, 110°14′28″ E, alt. 1100 m, 17 July 2014, beech forest, mixed leaf litter, humus, sifted, Peng, Song, Yu & Yan leg. (SHNU); 1♂, Guangxi, Jinxiu County, ’16 km’, 24°08′25″ N, 110°15′38″ E, alt. 960 m, 13 July 2014, beech forest, mixed leaf litter, humus, sifted, Peng, Song, Yu & Yan leg. (SHNU); 1♀, Guangxi, Jinxiu County, ’16 km’, 24°10′01″ N, 110°14′38″ E, alt. 1200 m, 10 July 2014, beech forest, mixed leaf litter, humus, sifted, Peng, Song, Yu & Yan leg. (SHNU); 1♀, Guangxi Prov., Jinxiu County, Yinshan P. R., 24°09′ N, 110°14′ E, alt. 1200 m, 23 July 2011, Peng leg. (SHNU).

**Description.** Measurements of males: BL: 10.83–11.72 mm, FL: 5.74–6.26 mm. HL: 1.52–1.61 mm, HW: 2.07–2.32 mm, EYL: 0.62–0.65 mm, TL: 0.68–0.81 mm, PL: 1.98–2.17 mm, PW: 1.67–1.86 mm, EL: 2.66–2.91 mm, EW: 2.69–2.85 mm. HW/HL: 1.36–1.44, TL/EYL: 1.10–1.25, PL/PW: 1.13–1.20, EL/EW: 0.99–1.02.

Measurements of females: BL: 9.76–12.27 mm, FL: 5.79–6.26 mm. HL: 1.58–1.67 mm, HW: 2.01–2.17 mm, EYL: 0.62–0.68 mm, TL: 0.74–0.77 mm, PL: 1.95–2.17 mm, PW: 1.71–1.89 mm, EL: 2.69–2.89 mm, EW: 2.88–2.91 mm. HW/HL: 1.27–1.35, TL/EYL: 1.09–1.19, PL/PW: 1.13–1.15, EL/EW: 0.93–1.00.

Head black, with faint metallic reflection, obscurely red around antennal sockets; pronotum, suture and scutellum brick red; elytra with posterior two-thirds black, with slight to distinct violaceous hue, black color extending onto hypomeron and almost reaching lateral margin, anterior third brick red, posterior margin reddish yellow; abdomen with segments III–V reddish, VI black with anterior margin narrowly reddish, VII black with posterior margin narrowly reddish yellow, VIII and IX yellow but becoming slightly darker toward posterior margin, X dark; antennae black, base of segment 1 and segment 2 reddish, segments 9–11 (rarely 10–11) creamy white, segment 8 sometimes black proximally, becoming paler off-white distally (Figure 2E); mandibles reddish brown, medial margin of mandible dark brown; mouthparts reddish to reddish yellow, maxillary and labial palpi brown; legs reddish brown.

Head (Figure 2G) 1.27–1.44 times as wide as long, rounded trapezoid, eyes slightly prominent, tempora narrowed toward neck in almost regular arc; surface of head with moderately dense and coarse punctation; frons impunctate; antennae with segments 4–10 markedly oblong.

Pronotum 1.13–1.20 times as long as wide, widest at about level of large lateral seta, narrowed toward base in shallow concave arc; surface as moderately densely and coarsely punctate as on head, with distinct impunctate midline, surface of pronotum between punctures without any microreticulation, shiny; scutellum with large but shallow punctation, surface between punctures with microsculpture of very fine transverse waves.

Elytra 0.93–1.02 times as long as wide, at shoulders distinctly narrower than at posterior margin, exceedingly densely, coarsely punctate.

Abdominal tergites III–V with basal transverse depression and a pair of short oblique basal accessory lines, punctation of abdominal tergites III–V shallow and sparse in depressions; posterior halves of abdominal tergites III–V and entire surface of remaining tergites with very fine and dense punctation.

Male. Protarsomeres 1–4 slightly dilated, heart-shaped; elytra with distinct and sharp, sinuous lateral keel; sternite VII with flat groove bearing dense and long yellowish brown setae; sternite VIII emarginate at middle of posterior margin. The specimens from the same locality showed slight variation in the aedeagus (Figure 6A,B,D–F), aedeagus with median lobe and paramere distinctly asymmetrical, paramere (Figure 6D–F) slender, shorter than median lobe. The variation of paramere is continuous, which should be considered as variability within the species.

Female. Protarsomeres 1–4 weakly dilated; elytra not keeled; tergite X (Figure 6C) nearly symmetrical with posterior margin emarginate at middle.

**Etymology.** The specific name is derived from its native country (China).

**Distribution.** China (Guangxi).

**Remarks.** The new species is similar to *H. kurbatovi* Schillhammer, 2009 from China (Sichuan) and *H. schoenmanni* Schillhammer, 2018 from China (Yunnan) and Vietnam, but can be distinguished from them by distal three antennomeres, rarely two creamy white, (distal four to five antennomeres creamy white in *H. kurbatovi* and *H. schoenmanni*); from *H. kurbatovi* also by bigger eyes with TL/EYL larger than 1.09–1.25 (1.35 in *H. kurbatovi*); and from *H. schoenmanni* also by the brick red pronotum (red to yellowish red with variable dark spot in *H. schoenmanni*).

Updated key to the species of *Hesperoschema* Scheerpeltz, 1965

The key is based on that by Schillhammer (2018) with three new species inserted.Entire forebody dark, without any reddish or yellowish markings……………………………………………………………………2–Forebody at least in part reddish, elytra at least obscurely reddish at base, posterior margin narrowly yellowish………………42.Forebody finely punctate, surface between punctures with microreticulation, matt ……………………………………***H. opacum***

China (Sichuan).–Forebody coarsely punctate, surface between punctures without microreticulation, glossy…………………………………33.Head and pronotum metallic violaceous; abdominal segments III–VI reddish ……………………………………***H. malaisei***

China (Yunnan), Myanmar.–Head and pronotum black to dark metallic olivaceous; abdominal segments III–V dark ………………………………………***H. sauteri***


China (Taiwan).4.Head sparsely punctate; profemur and coxae dark, strongly contrasting with pale prosternum …………………………***H. bii*** sp. nov.

China (Xizang).–Head densely punctate; profemur and coxa entirely reddish ………………………………………55.Pronotum and abdominal segments III–V dark………………***H. sauteri*** (colour variation)

China (Taiwan).–Pronotum at least partly and abdominal segments III–V reddish …………………………………66.Pronotum with variably extended dark markings, neck as dark as head…………………………………………***H. schoenmanni***

China (Yunnan); Vietnam.–Pronotum reddish, neck relatively brighter than head ……………………………………………77.Suture with slight to distinct violaceous hue …………***H. xuwangi*** sp. nov.

China (Sichuan).–Suture reddish yellow……………………………………………88.TL/EYL: 1.35; distal five antennomeres creamy white …………………………………***H. kurbatovi***

China (Sichuan).–TL/EYL: 1.09–1.25; distal two to four antennomeres creamy white …………………………………………***H. sinicum*** sp. nov.

China (Guangxi).

## 4. Discussion

*Hesperoschema* species are colorful rove beetles. The coloration of antennae is always bicolored in the genus: the distal two to five antennomeres are creamy white, while the remaining antennomeres are blackish. In Anisolinina, the number of the white distal antennomeres is usually stable in species and may be regarded as a fast way of identification. Yet, some cases of variation have been found in *H. sinicum* and. *H. schoenmanni*. Therefore, the identification based on the coloration of antennomeres should be applied carefully in *Hesperoschema*.

The biological knowledge of the genus is almost completely unknown. *Hesperoschema* species are usually quite rare. They are not easily collected and can only be found in primary forests. Fungi on logs are the typical habitat for *Hesperoschema* species, and occasionally, they were spotted crawling on the surface of animal corpses. The hunting behavior has not been observed, yet their feeding habits can be inferred by the structures of the mouth parts. The mandibles are slender and elongate and cannot afford a strong biting force. Therefore, they very likely feed on beetle larvae and maggots with soft bodies, just like their relatives in *Hesperosoma* [8,9].

The species of *Hesperoschema* are alert and active with excellent visual sense. Detecting the approach of the collector, they always run fast and sometimes try to fly away. These observations may make an impression that *Hesperoschema* species have good dispersal ability. Yet, based on the material published so far, the distribution of each species is limited in small area. No overlapping distribution has been found, though the distributional localities of some species are not very far away. This might be due to their rarity and limited samples. A distributional map based on all known specimens is given in Figure 7. The discoveries of *H. bii* and *H. xuwangi* expand the westernmost and northernmost distributional border of the genus, respectively. Six of the eight *Hesperoschema* species, including two species mentioned above, are scattered across the mountains of East Himalaya and its eastern extensions. *Hesperoschema sauteri* is the easternmost species in the genus distributed in Taiwan. There is a huge distributional vacuum in South China between *H. sauteri* and the other species, though the discovery of *H. sinicum* partially fills the vacuum. Such geographic distributional pattern implies that East Himalaya and its eastern extensions are the biodiversity hot spots of the genus. More new species may be possibly found from there in the future, and the westernmost and northernmost distributional border may be expanded to Nepal and South Ningxia Province, China.

## Figures and Tables

**Figure 1 insects-13-00060-f001:**
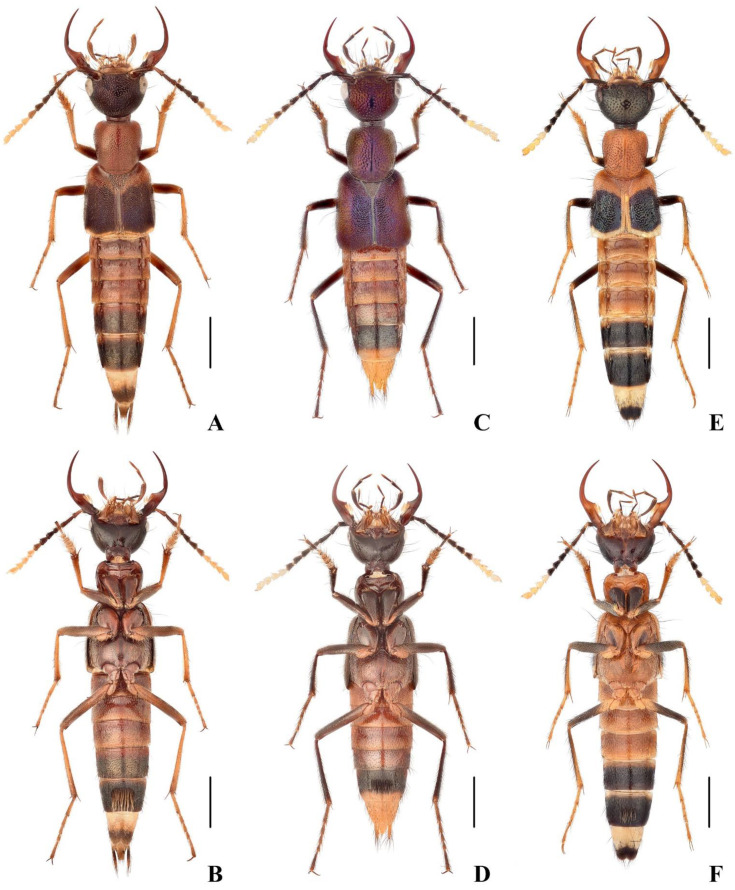
(**A**–**F**). Adult habitus of *Hesperoschema*. (**A**,**B**) *H. xuwangi*; (**C**,**D**) *H. malaisei*; (**E**,**F**) *H. bii*. Scale bars = 2 mm.

**Figure 2 insects-13-00060-f002:**
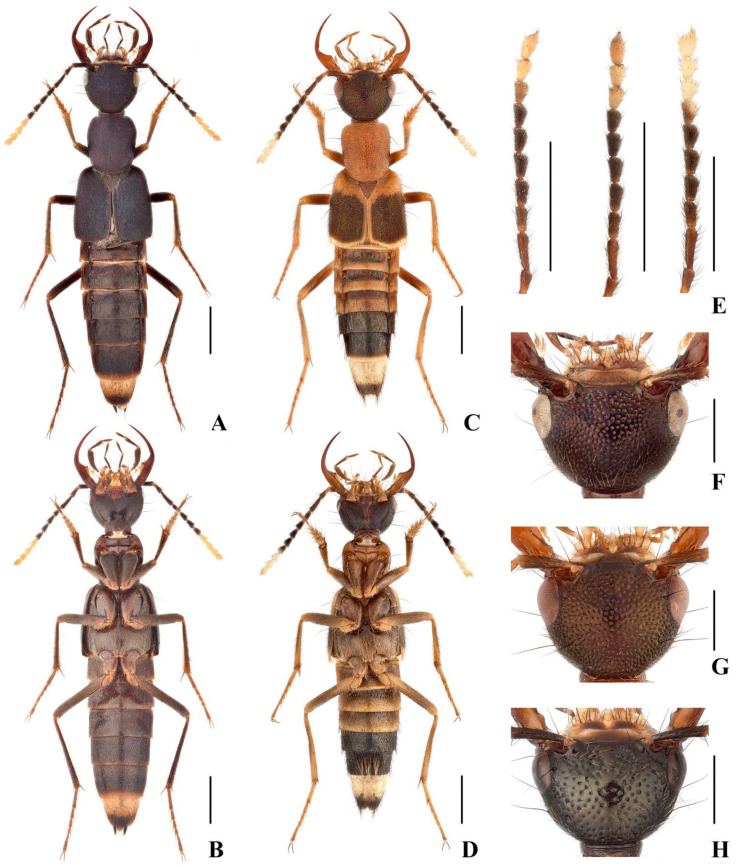
(**A**–**D**) Adult of *Hesperoschema*. (**A**,**B**) Adult habitus of *H. opacum*; (**C**,**D**) adult habitus of *H. sinicum*; (**E**) antennae of *H. sinicum*; (**F**–**H**)**.** Head of *Hesperoschema*. (**F**) *H. xuwangi*; (**G**) *H. sinicum*; (**H**) *H. bii*. Scale bars: 2 mm (**A**–**D**), 1 mm (**E**–**H**).

**Figure 3 insects-13-00060-f003:**
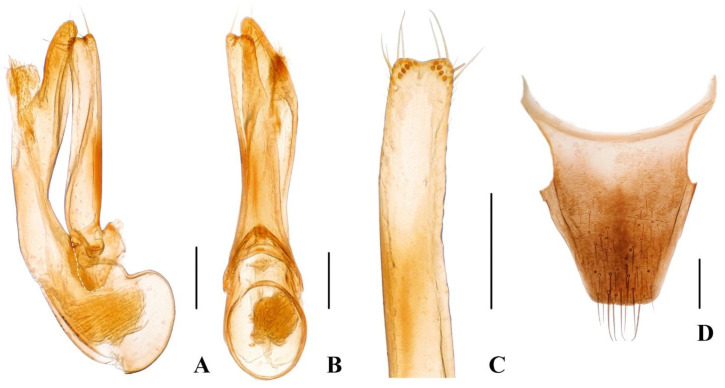
(**A**–**D**) Diagnostic characteristics of *Hesperoschema*. (**A**–**C**) *Hesperoschema xuwangi*. (**A**,**B**) Aedeagus, lateral (**A**) and ventral (**B**) views; (**C**) paramere; (**D**) female abdominal tergite X of *H. opacum*. Scale bars = 0.2 mm.

**Figure 4 insects-13-00060-f004:**
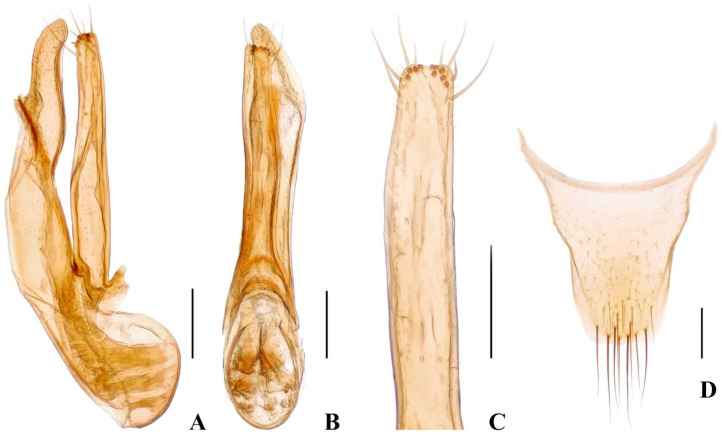
(**A**–**D**) *Hesperoschema malaisei*. (**A**–**C**) Aedeagus, lateral (**A**) and ventral (**B**) views; (**C**) paramere; (**D**) female abdominal tergite X. Scale bars = 0.2 mm.

**Figure 5 insects-13-00060-f005:**
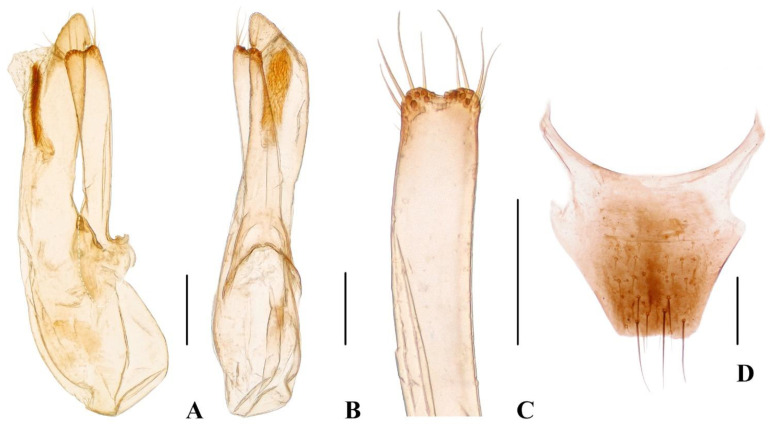
(**A**–**D**) *Hesperoschema bii*. (**A**–**C**) Aedeagus, lateral (**A**) and ventral (**B**) views; (**C**) paramere; (**D**) female abdominal tergite X. Scale bars = 0.2 mm.

**Figure 6 insects-13-00060-f006:**
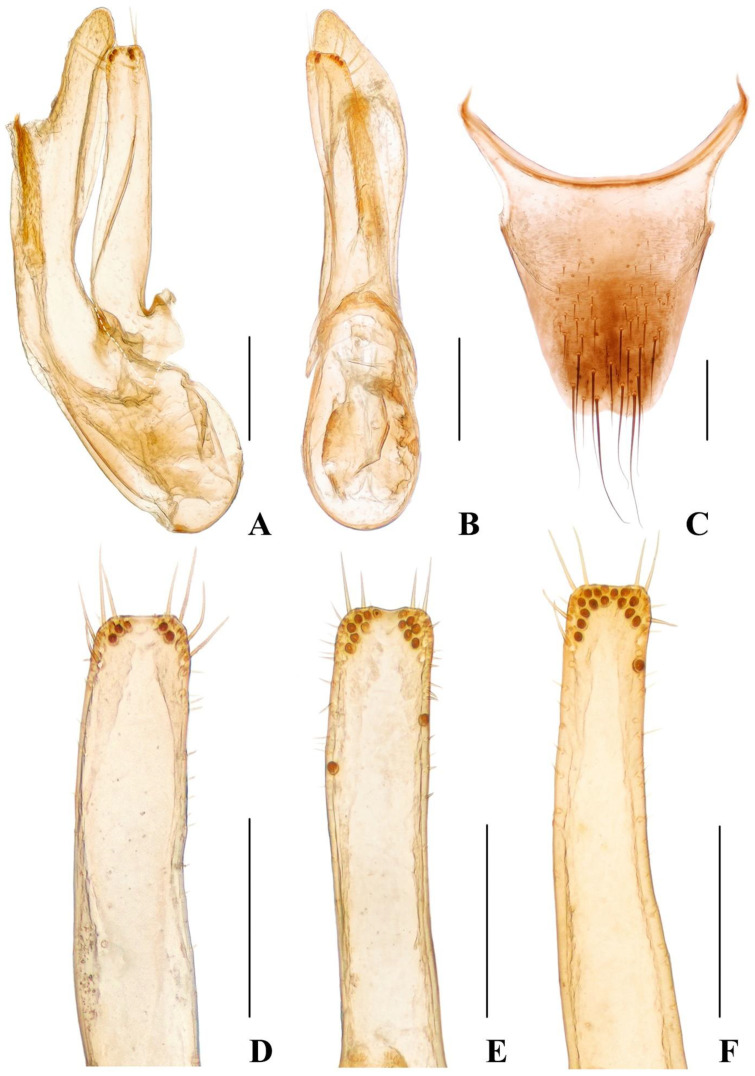
(**A**–**D**) *Hesperoschema sinicum*. (**A**,**B**) Aedeagus, lateral (**A**) and ventral (**B**) views; (**D**–**F**) paramere; (**C**) female abdominal tergite X. Scale bars = 0.2 mm.

**Figure 7 insects-13-00060-f007:**
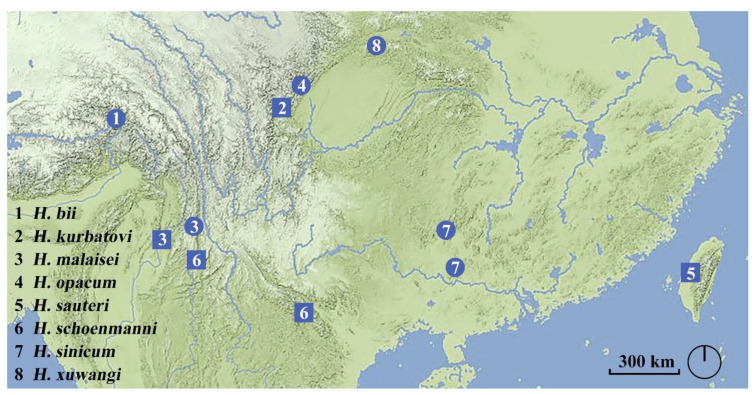
Distribution of all species of *Hesperoschema*. Circle, Localities of examined specimens in this paper; square, localities of specimens listed in previous papers.

## Data Availability

No new data were created or analyzed in this study. Data sharing is not applicable to this article.

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
