# Peer review of "Three New Species and One New Record of Hesperoschema (Coleoptera: Staphylinidae: Staphylininae) from China†"

_insects, 2022, doi:10.3390/insects13010060_

Round 1

Reviewer 1 Report

My remarks and corrections of some places in the manuscript:

Page 1, lines 9, 10, 13, 16, 32. “5 species…”, “4 known…”, 1 known…”, “3 new…”, “to 8.”

I recommend to replace numeric symbols into capital letters: “five species…”, “four known…”, etc.

Page 1, line 18: “….from China (Guangxi), is described.” should be changed to “….from China (Guangxi) are described.”

Page 2, line 37: “Anisolinus” should be in itallic

Page 2, lines 46, 47: “photos” should be changed to “photographs”

Page 2, line 50: “The type specimens treated in this study are deposited in the following public collections…”. Since the material from only one museum are used in the present study, I recommend to change this place in the text as follows: “The type specimens treated in the study are deposite in SHNU (Department of Biology, Shanghai Normal University, P. R. China).”

Page 2, line 58 “length of eye”. Is it longitudinal length of eye?

Page 3, line 74 (here and in similar places below): “mm”. I recommend to remove all similar places in the description and to made additional sentence in the “Materials and Methods section” “all measurements are given in millimeters.”:

Page 3, line 77 “Head black with…”. The punctuation in the text of MS should be corrected. In this case to add a comma between “black” and “with”.

Page 3, line 87 “Head (Fig. 3F) 1.49 times as wide as long…”. I recommend reduced similar numeric values to one decimal place, in this case it can be “about 1.5 times as wide as long…”.

Page 5, line 96 “Pronotum 1.10 times as long as wide…”. See above, “…1.1 times”. Or, may be better: “Pronotum slightly longer than wide…”

Page 7, line 127 and line 129: “Scheerlpeltz…” should be changed to “Scheerpeltz…”

Page 7 line 129 Scheerpeltz 1965 missing in the list of references

Page 7 line 129 Hayashi 2003 missing in the list of references

Page 7 line 141 (here and in all similar places of MS): “Measurements of male…”. As I understand, these measurements were made for all males, so it would be better to change it as follows: “Measurements of males…”

Page 7 line 145 (here and in all similar places of MS): “Measurements of female…”. See above.

Page 7 line 154: “Female. Protarsomeres 1-4 female…” To remove word “female”.

Page 9, line 186: “Head (Fig. 3H) 1.53 times as wide as long…”. See above.

Page 9, line 190: “Pronotum 1.09 times as long as wide…”. See above.

Page 11, line 265: “Head (Fig. 3G) 1.27-1.44 times as wide as long…”. See above.

Page 11, line 269: “Pronotum 1.13-1.20 times as long as wide…”. See above.

Page 12, line 274: “Elytra 0.93-1.2 times as long as wide…”. See above.

Page 12, line 279 (here and in all text): “….punctuation” should be changed to “…punctation”

Page 14, line 305, 306: “…fore body…”, “Fore body…”. It would be better if to combine these words into one “forebody”.

Page 15, line 363: “…the hereto known…” I not understand all this sentence.

Page 16, line 388 “Hayashi…2002”. This reference missing in the text or may be confused with Hayashi 2003

Author Response

I have received the review comments and have been revised the manuscript. But I still hope to keep two digits, so that it can be better compared with the data in the previous paper.

Reviewer 2 Report

The manuscript titled “On The Genus Hesperoschema (Coleoptera: Staphylinidae: Staphylininine) of China” is a continuation of taxonomic research concerning Chinese members of Anisolinina. The present research involves the small genus Hesperoschema (Staphylinnidae), currently including only a few species on a global scale. These are very interesting rove-beetles, only inhabiting the primary forests in Southern Asia Region. The specific environmental preferences and quite limited distribution of these beetles, make them for entomologists a very interesting object of taxonomic and zoogeographic study. Thus, new information presented in this work is very needed and valuable. They constitute another important step forward in expanding knowledge about taxonomy and the distribution of that subtribe. The manuscript contains all the necessary elements, typical for studies in the field of alpha taxonomy. They have been exhaustively developed, in accordance with current standards applicable in scientific studies of this type. Descriptions of morphology of discussed taxa are exhaustive, transparent, with accompanying good quality, viewing photographs.

The added value is also the well-constructed, sensible key for distinguishing all known members of Hesperoschema, that also includes the newly described taxa. I also do not have any objections to a linguistic job.

Therefore, the manuscript should be published in the Insects journal, but after taking into account the following three suggestions/correction (minor revision).

  1. I suggest that the map distribution (Figure 1) of known Hesperoschema species should be move after Discussion (before Author Contributions), and thus just after the zoogeographic analysis carried out in that chapter. Logical order should be as follows: the first - description of the three new species and complementing the description of two already known species, the second - their geographical distribution with the map, not vice versa as it is now.

  1. In the first line on page 11, just under the species name - Hesperoschema sinicum sp. nov. - currently, it is - 6A-F, and should be – 7A-F.

  1. I suggest also to change the Discussion chapter name to Coloration, ecological preferences, and distribution. In the presented version, this chapter does not contain any elements of scientific discussion, but only synthetic information on the above-mentioned three issues.

Author Response

I have received the review comments and have been revised the manuscript.

Reviewer 3 Report

The authors present an update to the taxonomy of rove beetle genus Hesperoschema. These beetles are rarely collected and are likely indicators of high quality forest habitats, so knowledge of their biodiversity and the development of identification resources are of great importance. The manuscript is generally well-written, the illustrations are of very high quality and the evidence for new species is well presented. I have mostly corrections to the clarity of the manuscript, a few incorrect figure references, and one other issue. The authors suggest that the species of the genus are very local, based on the known distribution. However, this is simply an artefact of very few known specimens, their rarity, and potentially, the difficulty to access a variety of good forest habitats. The species of the genus are distributed in the classic biodiversity regions of China (at least where Staphylininae are concerned) and they are quite different from locally endemic, flightless rove beetles where new species are found in each local cluster of mountains.

Otherwise, this is an excellent contribution to the study of Staphylinidae and will make a valuable contribution to the special issue.

Author Response

(The authors gave the same response as above.)
